# Rescheduling Out-of-Gauge Trains with Speed Restrictions and Temporal Blockades on the Opposite-Direction Track

Zhengwen Liao

State Key Laboratory of Rail Traffic Control and Safety, Beijing Jiaotong University, Beijing 100044, China; zwliao@bjtu.edu.cn; Tel.: +86-10-51688547

**Abstract:** Out-of-gauge trains are trains with loading freight that exceeds the loading limitation border. Considering collision avoidance, the out-of-gauge trains have speed restriction of their own, and the trains running on the parallel track. Therefore, it is necessary to execute a train rescheduling procedure to rearrange the train paths of the out-of-gauge trains and the affected trains based on the fundamental timetable. For rescheduling the timetable, considering the blockades and the speed restrictions caused by the out-of-gauge trains, this paper proposed a time-space-state network representation for describing the out-of-gauge train rescheduling problem. A novel concept, speed allowance, is introduced to describe the train speed restriction due to the out-of-gauge trains. An integer programming model based on the time-space network is proposed to minimize the total train delay when running the out-of-gauge trains. The model can be solved by the rolling-time horizon approach for reducing computational time. A numerical example is conducted based on the conventional railway in China, demonstrating the solution performance of the model and the practical use of the methodology. Gurobi solver cannot obtain an optimal solution within 1 h when the planning-time horizon is greater than 120 min. With the rolling-time horizon approach, the rescheduled timetable can be obtained within 124 s for the 300 min planning-time horizon using 180 min rolling-time window.

**Keywords:** railway rescheduling; train dispatching; out-of-gauge train; optimization; integer programming

**MSC:** 90B06





## 1. Introduction

### 1.1. Background

Train timetables specify the arrival and departure times for trains at each station along their route. In most railway transportation systems, railway infrastructure managers generate train timetables in different planning phases for specific purposes. The fundamental timetable is a predefined long-term plan, usually scheduled a few months before execution. However, the timetables executed on certain days might vary from the fundamental timetable, especially in the daily timetable; some trains in the fundamental train timetable are canceled due to the decreasing passenger or freight flow, while some trains are temporally decided to run additionally due to exceptional demand (e.g., increasing passenger flow, extra wagon circulation task). These timetable amendment decisions are made to generate a daily timetable by dispatchers one or a few days before the timetable is executed. Traffic rescheduling focuses on network capacity and the need for the infrastructure manager (IM) to revise the timetable and allocate track resources for the affected trains, in order to minimize delays [1]. When the timetable is being implemented, train dispatchers monitor the train traffic and make necessary real-time rescheduling decisions to recover the train trajectory (time-space path) of the daily timetable under disturbances. The planning phases of the train timetable are shown in Figure 1. According to Corman and Meng [2],

train timetables, as tactical plans, are programmed and updated every year or every season (offline) to define routes and schedules of trains. In daily train operations, various sources of perturbations may influence train running times, as well as dwell and departing events, thus causing primary delays to the planned train schedule. In the state-of-the-art of train scheduling, many researchers focus on the real-time train rescheduling approach, which can be referred to as the literature review [3]. However, in this paper, we study the daily rescheduling before the timetable is implemented.

### Planning phases of train timetabling

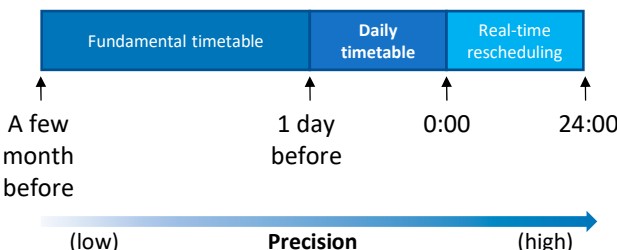

**Figure 1.** Planning phases of train timetables.

Among the timetable amendments in daily timetable rescheduling, one essential task is rescheduling the out-of-gauge trains. Typically, the size and shape of the freight being transported by trains are limited to a particular train gauge (the maximum border of the vertical cross-section). The cross-section of train gauge and out-of-gauge train loading can be referred to as UIC loading guidelines [4]. However, some freights (e.g., oversized machines) are impossible to load on a train within the loading gauges. An out-of-gauge train is a train that exceeds the different levels of standard loading gauges.

Although the out-of-gauge trains exceed the standard loading gauge, they can still be run but with speed and pass-by restrictions, such as:

- Out-of-gauge trains should run at a lower speed to avoid possible collisions due to the vibration;
- The train pass-by (two trains running on opposite direction tracks meets at segments between two adjacent stations) is forbidden for some types of trains;
- The speed reduction of the parallel track is applied when the segment is running an out-of-gauge train;
- Station tracks that can accommodate out-of-gauge trains are limited. Therefore, the meeting and overtaking stations of out-of-gauge trains should be carefully decided.

For the reasons above, when a freight train path in the fundamental timetable is decided to run an out-of-gauge train, this train path and the corresponding influenced train paths should be rescheduled. In particular, due to the speed reduction of the out-of-gauge train, trains running in the identical direction might suffer from delay propagation, while trains running in the opposite direction might also be influenced by the out-of-gauge train, due to the pass-by restriction. The impact of the scheduled out-of-gauge trains on the other trains, especially the pass-by restriction to the opposite direction train, makes it a significant challenge to make a daily timetable. Therefore, a fast-rescheduling algorithm would help create a daily timetable from a fundamental timetable, reduce timetable schedulers' labor intensity, and accelerate the daily timetable publishing procedure.

Modeling the train rescheduling problem of out-of-gauge trains is difficult, as the typical minimum headway constraint between two successive trains can hardly describe the complicated pass-by restriction of the out-of-gauge trains. In this paper, we formally define the out-of-gauge train rescheduling problem as creating the daily timetable by minimizing the weighted train delays compared with the fundamental timetable considering the speed restriction and temporal parallel track blockade constraints. Concerning the speed restrictions or track temporal blockades caused by out-of-gauge trains, we propose a novel concept named 'speed allowance' based on the time-space network representation of the

timetable. This approach can implicitly describe the specific speed restrictions or temporal blockades to the trains running on the parallel opposite direction track caused by the out-of-gauge trains. A network flow-based integer programming model, and an associated rolling time horizon solution method, are applied to solve the out-of-gauge train rescheduling problem. Case studies involving an artificially generated timetable and a case study for a practical timetable are conducted to demonstrate the solution quality, efficiency, and numerical patterns of the timetable with the out-of-gauge trains.

The remainder of this paper is organized as follows. Section 1.2 is the literature review that summarizes the recent study concerning train rescheduling, especially with speed variation consideration, followed by a contribution statement of this paper in Section 1.3. Section 2 analyzes the generic speed restriction regulation and formally defines the train rescheduling problem with speed restriction due to the out-of-gauge trains. Section 3 proposes a mathematical model and an associated rolling-time horizon algorithm of the train rescheduling problem based on a time-space network representation with 'speed allowance' constraints, which describe the speed restriction caused by out-of-gauge trains. Section 4 provides the case study of rescheduling out-of-gauge trains using the proposed approach to demonstrate the solution efficiency, quality, and performance of the timetable derived from out-of-gauge train rescheduling.

### 1.2. Literature Review

The train rescheduling problem is widely studied as a discrete combinatorial optimization problem. While the train timetabling problem is considered an offline optimization problem, the train rescheduling problem is often regarded as a real-time decision-making problem. Train rescheduling problem specifies the new arrival and departure times, local or global routes, or new train services under disturbance or disruptions. The recent train rescheduling survey can be referred to as Cacchiani et al. [3], Wen et al. [5], and Jusup et al. [6]. Dong et al. [7] further concludes that the integration of train rescheduling and train control problem, where the train dynamics are described more thoroughly.

In macroscopic models, the train speed is often modeled as a fixed value or a predefined range. Zhan et al. [8] applied a space-time-speed network to model the energy-efficient train rescheduling with train trajectory optimization, where the running time of trains are determined by a train trajectory model. Wang et al. [9] modeled the train optimization of train rescheduling and planned the rolling stock circulation for metro disruption, where the running time is applied as a min and max range. Hong et al. [10] modeled a train rescheduling problem considering passenger reassignment, where the running time is subject to a given range with minimum and maximum speed. Zhang et al. [11] specified different minimum and maximum running times according to the speed limit of specified train types. The train running times under disruption are obtained from real-time information instead of being predetermined. Zhan et al. [12] applied 'drive arc' to represent the train running in a segment between two stations. Principally, the drive arcs for a train in a segment are within a limited set, implying that the running time is a predetermined subject to a min-max range. Meng et al. [13] studied the station track assignment problem under a train rescheduling optimization framework, where the running time is set to be longer than the minimum running time. Josyula et al. [14] proposed a parallel computing technic for a multi-objective train rescheduling problem, where the train speeds are variables with a speed limitation range. Long et al. [15] regarded the train running speed in the segment as a variable under temporary speed restriction. The speed restriction is determined by the scenario (i.e., normal and disruption). Hou et al. [16] proposed an energy-saving metro train timetabling problem considering ATO profiles. The train running time is determined by train operation levels with different train trajectories. Xu et al. [17] proposed a minimum running-time restriction under a quasi-moving block system. The train running in a blockade segment speed adjustment is taken into consideration. Hong et al. [18] applied a time-space network to model the variation in train running speeds. Altazin et al. [19] proposed a rescheduling model for stop-skipping in

dense railway systems and an optimization simulation framework. The running time is also regarded as a min-max range. The model describes the resource occupation constraint on the track circuit level. The running times on track circuits are defined within min-max ranges. Cavone et al. [20] introduced an MPC-based rescheduling algorithm for solving the train rescheduling problem in a cross-granularity manner. The actual train running time is collected at each step, and the rescheduling decision is made based on the updated train running time. The running time is determined in a fine-tuning module where the train running time is calculated considering energy efficiency and robustness. Xie et al. [21] determined around a 5% to 7% running-time buffer when scheduling trains, and the actual train running time is determined both by energy and passenger factors. Yuan et al. [22] studied a train timetable, rolling-stock assignment, and short-turning strategy problem. The running time at each segment and dwelling time at each station are predetermined. Liao et al. [23] modeled the train running time, dependent on the stop-pattern decision. The acceleration and deceleration additional running time is considered. In the studies above, the train running time between two adjacent stations only depends on the technical speed of the train and with no correlation with other scheduled train paths. In other words, the running speeds are regarded as independent variables, and not directly interfered with by other trains.

However, the train running time in segments is impacted by the train speed variation, influenced by the temporal speed restriction due to environmental conditions (e.g., bad weather). Some papers study the train speed variations in the integrated-optimization problem of train rescheduling and maintenance time window arrangement. Luan et al. [24] first studied the integration of train rescheduling and preventive maintenance time slot planning. The influence of the maintenance time slot on the train path is modeled as a virtual train. Trains are forbidden to have conflicts with the maintenance slots. Zhang et al. [25] further addressed an improved train rescheduling problem introduced in the Informs RAS problem solving competition 2016 [26], considering the impact of the maintenance time window on the train running speed. Zhang et al. [27] applied a layered space-time network to model the joint-train scheduling problem. The impact of maintenance on train running time is modeled as the incompatible nodes. Zhang et al. [28] applied a Lagrangian relaxation decomposition approach to tackle the train scheduling problem under emergency maintenance. The train running time is restricted to a minimum value if the train runs on the maintenance section, in the opposite direction of the maintaining section, or after corresponding maintenance tasks. Wang et al. [29] considered the impact of overnight maintenance planning on the overnight train for high-speed railways. The model can optimize both the layout of the overnight maintenance time slot and the train schedules of overnight trains. Zhang et al. [30] proposed a heuristic procedure dynamically updating the available time windows for each train to solve the train timetabling, platforming, and railway network maintenance scheduling decision problem. During the scheduled infrastructure maintenance duration, the train on the parallel track should run at a lower speed for construction safety reasons. The running time between two adjacent stations is affected by the technical speed and the time-space relationship between the train path and the maintenance time window. If the train path overlaps with the maintenance time window, the speed restriction of the train on the associated segment should be complied with.

The uncertainty factors are commonly considered in train rescheduling models. Peng et al. [31] studied traffic management optimization under uncertain temporary speed restrictions. Liu et al. [32] applied an alternating direction method of multipliers (ADMM) combined with the model predictive control (MPC) approach to solve the timetabling and platforming problem in case of uncertain perturbation. Zhang et al. [33] applied a multistage decision approach for optimizing the train timetable rescheduling under uncertain disruptions. Liebhold et al. [34] introduced a dynamic onboard tuning method of energy-efficient speed profiles after the real-time train rescheduling process under a fixed block-signaling system for mixed traffic. Gao and Vansteenwegen [35]

proposed a mixed integer linear program to optimize the response to partial blockages, considering the use of reversible tracks.

The characteristic comparison of the literature concerning train scheduling and rescheduling can be referred to in Table 1.

**Table 1.** Recent literature of train scheduling and rescheduling.

| Literature | Model Category | Solution Method | Speed Variant | Station Track Assignment | Problem Scale |
|---|---|---|---|---|---|
| Zhan et al. [8] | Space-time-speed network | ADMM | Y (trajectory optimization) | Y (number limit) | 14 stations, 14 trains |
| Wang et al. [9] | MIP | Two-stage heuristic | Y (min-max range) | N | 21 stations, 20 trains |
| Hong et al. [10] | MIP | CPLEX | Y (min-max range) | Y | 8 stations, 20 trains |
| Josyula et al. [14] | Not applicable | Parallel computing (depth-first search) | Y (min-max range) | Y | 59 sections, 5 h 20 min planning horizon |
| Long et al. [15] | MIP | CPLEX | Y (min-max range) | Y | 2 stations, 2 trains |
| Altazin et al. [19] | Event-activity network | Optimization + simulation | Y (min-max range) | N | 13 train lines |
| Cavone et al. [20] | MIP | Model predictive control | Y (micro-macro interaction) | Y | 75 min time horizon |
| Xie et al. [21] | MIP | Genetic algorithm | Y (buffer reservation) | N | 8 stations, 10 trains |
| Luan et al. [24] | Time-space network | Lagrangian relaxation | Y (speed restriction) | Y | 5 stations, 31 trains |
| Peng et al. [31] | MIP | Rolling time horizon | Y | Y | 5 stations, 30 trains |
| Liu et al. [32] | Event-activity network | ADMM + MPC | Y (min-max range) | Y | 23 stations |
| Liao (this paper) | Time-space network | Rolling time horizon | Y (out-of-gauge regulations) | Y | 10 stations, 44 trains |

Concerning the out-of-gauge train operation, Zhang et al. [36] proposed an optimal route generation method for railway out-of-gauge freight, where safety (i.e., the gap clearance) and economic factors are considered in the optimization model. Based on this model, Zhang et al. [37] further considered the railway capacity losses and transportation costs, as well as the gauge modification, to generate more comprehensive route decisions. Zhang et al. [38] investigated the optimal location, length, and the number of non-crossing block sections for out-of-gauge trains to reduce railway capacity loss by applying a cellular automata simulation-based method. Ju et al. [39] studied the impact factors of the classification of out-of-gauge limits and proposed an associated mathematical model for classifying the gauge limits. In general, the existing research on out-of-gauge operation focuses on the safety aspects (e.g., determining a safe route for the out-of-gauge trains). Some pieces of the literature consider the macroscopic impact (i.e., capacity loss) when running out-of-gauge trains.

However, to our best knowledge, the existing research has not discussed the impact of out-of-gauge trains on its own speed, as well as the related trains. Due to the following characteristics, the problem deserves to be studied. Firstly, the problem is an offline problem; unlike the typical real-time train rescheduling problem, the solution efficiency requirement is relatively low. Thus, more sophisticated solution strategies can be applied. Secondly, the schedule of out-of-gauge trains has leveled and categorized speed restrictions. Unlike the fixed speed restriction under infrastructure maintenance, the out-of-gauge train can select a speed restriction to run among a predetermined set. The candidate speed-restriction schemes impact the trains running on the parallel opposite direction track differently. In general, the out-of-gauge train scheduling problem proposes an interesting topic of balancing the profit between out-of-gauge trains and normal trains by rearranging the departure time, station-track utilization, and overtaking points.

### *1.3. Contribution Statements*

To our best knowledge, although many publications study the train rescheduling problem considering speed variation (e.g., speed reduction caused by bad weather), very few studies (only several train rescheduling models considering infrastructure maintenance works) tackle the multi-level speed restrictions or blockades interfered by the out-of-gauge train paths, which are the decision variables themselves. This paper bridges the knowledge gap of the out-of-gauge train rescheduling problem by addressing the following issues.

1.  We propose a novel concept, namely, 'speed allowance', based on the time-space network for describing the complicated train running speed interference of out-of-gauge trains. Speed allowance can denote the different and complex occupation allowance and speed-level allowance.
2.  The rolling-time horizon approach, with multiple resource occupations and interferences, is applied to solve the problem efficiently.

## **2. Problem Description**

### *2.1. Scheduling Limitation for Out-of-Gauge Trains*

The classification and the scheduling regulation of out-of-gauge trains vary from country to country. However, the classes of out-of-gauge trains can be categorized by the difference between the actual size of the out-of-gauge train and the standard gauge in general. UIC loading guidelines [4] defines four sizes of out-of-gauge trains, namely, GA, GB, GB+, and GC.

The standard train loading gauge has abundant buffer space considering the vibrations and offsets during train running with normal speed. However, the buffer space can be compromised. The freight loading on a wagon can be out of the normal clearance but not exceeding the extended clearance. If a train loads out-of-gauge freight, the train becomes an out-of-gauge train. The out-of-gauge trains can be categorized into different levels according to the distance beyond the normal clearance. The out-of-gauge trains have to reduce the running speed to avoid collision with the devices, structures, and other trains running (typically in the opposite direction) on the parallel track. For example, the out-of-gauge train speed restriction in China can referred to in Table 2.

**Table 2.** Out-of-gauge train speed restriction in China Railway (TG/HY 106-2016) [40].

| Train Speed of Opposite Direction Train (km/h) | Out-of-Gauge Distance (mm) | Train Speed Limitation of the Out-of-Gauge Train (km/h) |
|:---:|:---:|:---:|
| | >350 | (no limitation) |
| ≤120 | 300~350 | 30 |
| | <300 | 0 (blockade) |
| | >450 | (no limitation) |
| 120~160 | 400~450 | 30 |
| | 400 | 0 |
| >160 | / | 0 (blockade) |

### *2.2. Rescheduling Problem Considering the Pass-by Limitation of Out-of-Gauge Train*

Running the out-of-gauge train might result in a speed restriction with the out-of-gauge train itself or the associated trains running on the parallel track. The speed restriction makes the fundamental timetable infeasible, and the interfered trains might suffer from consecutive delays caused by train reductions and temporal blockades. Therefore, the traffic manager reschedules the fundamental timetable to make it feasible before the operation day. This timetable amendment task derives a new timetable, namely, the daily timetable. The daily timetable will be issued to the corresponding managers in stations, locomotive and wagon depots, as well as other related departments, for better preparation to recover traffic with potential delays.

When the traffic manager plans the daily timetable, they must amend it according to the received out-of-gauge train-loading plan. Generally, several timetable constraints must be followed when scheduling the out-of-gauge train, resulting in the following consequences, as illustrated in Figure 2.

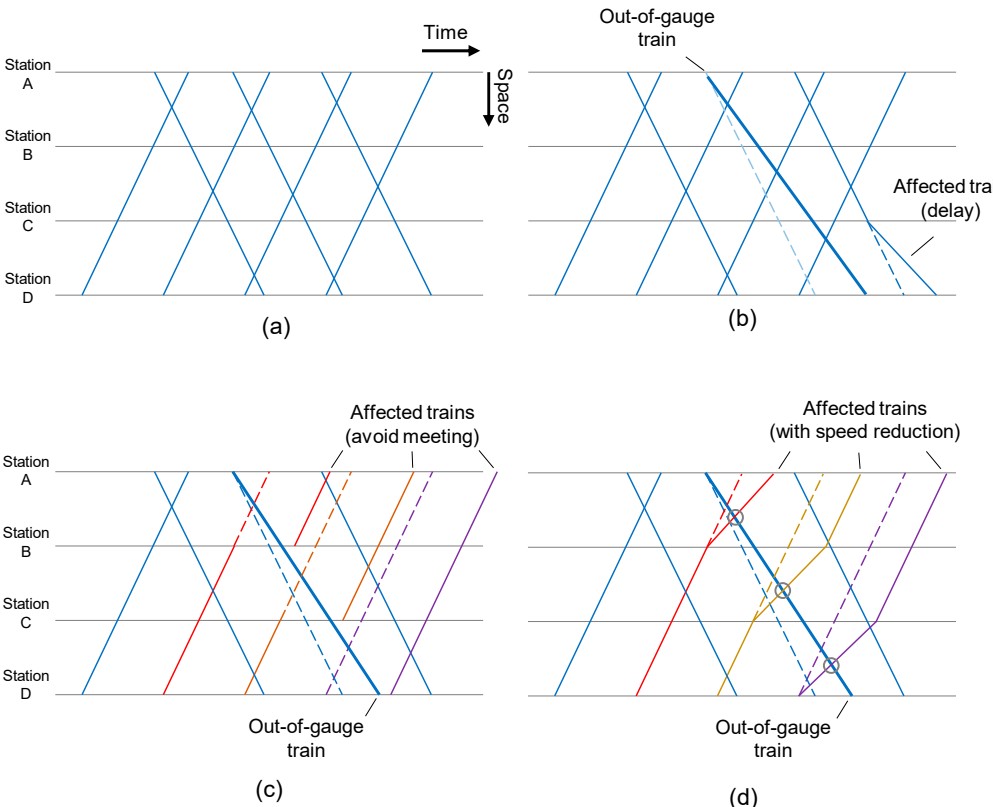

**Figure 2.** Illustration examples of the rescheduling of out-of-gauge trains. (**a**) Original timetable; (**b**) Identical direction impact; (**c**) Opposite direction impact (by avoid meetings); (**d**) Opposite direction impact (by speed restriction).

(1) Speed reduction of the out-of-gauge trains. As the analysis in Section 2.1, the out-of-gauge trains reduce their speed to a very low level to avoid the possible collision with the trains running on the parallel track in the opposite direction. In this case, the trains running on the parallel track in the opposite direction maintain their original speed. Therefore, the out-of-gauge train running at a restricted speed only results in train delays in one direction.

(2) Temporal blockade of the opposite track. For the highest level of out-of-gauge distance, the parallel track should keep vacant from the beginning to the end of the out-of-gauge train running through the segment. Therefore, the out-of-gauge train not only causes delays in its direction (due to speed restrictions) but also causes train delays in the opposite direction, as the associated trains need to rearrange the extra waiting time in stations with the out-of-gauge train. Note that, although the trains in the opposite direction cannot enter the segment during the out-of-gauge train running inside, the trains following the out-of-gauge train can enter the segment consecutively, as long as the headway constraint is satisfied.

(3) Speed reduction of the opposite trains. Note that, the possible collision depends on the speed of the out-of-gauge train and the train running on the parallel track. For the out-of-gauge train, we might reduce the running speed of the train running on the parallel track to avoid the temporal blockade. This approach can be beneficial to maintain the continuity of traffic, however at a lower speed, which is important to reduce capacity loss, especially in busy corridors.

From the analysis above, for the out-of-gauge train, many possible rescheduling strategies can make the timetable compatible with the speed restriction constraints of the out-of-gauge train. However, the possible application of rescheduling strategies might result in various negative impacts (e.g., different levels of train delays) for different traffic scenarios. This combinatorial effect yells an optimization problem that optimizes the train orders and running speeds, as well as the overtaking decision of the out-of-gauge train and the influenced trains to maximally maintain the original schedule (i.e., reduce the train delay caused by the scheduling limitation of out-of-gauge trains).

## 3. Mathematical Model

### 3.1. Notations

The notations used in the following description is displayed in Table 3.

**Table 3.** Notations.

| Notation | Description |
|---|---|
| Elements and collections for time-space network | |
| $t$ | Discrete time index |
| $s$ | Station element |
| $e(s_1, s_2)$ | Segment element between station $s_1$ and $s_2$ |
| $k$ | Station track element |
| $f$ | Train element |
| $v$ | Generic node of the time-space network |
| $V_f$ | Node set for train $f$ |
| $v_f^O$ | Origin node of train $f$ |
| $v_f^S$ | Sink node of train $f$ |
| $v_f(s, t, \mathrm{A})$ | Arrival time-space node for train $f$ in station $s$ at moment $t$ |
| $v_f(s, t, \mathrm{D})$ | Departure time-space node for train $f$ in station $s$ at moment $t$ |
| $v_f(s, t, k)$ | Track time-space node for train $f$ in station $s$ at moment $t$ on track $k$ |
| $t(v)$ | The time index of time-space node $v$ |
| $a_f$ | Generic time-space arc for train $f$ |
| $a_f(v, v')$ | Time-space arc for train $f$ from node $v$ to node $v'$ |
| $A_f$ | Time-space arc set for train $f$ |
| $A_v^+$ | Time-space arc set that entering node $v$ |
| $A_f^-$ | Time-space arc set that leaving node $v$ |
| $r$ | Time-space resource |
| $R$ | Global time-space resource set |
| $R_{a_f}$ | The time-space resource set that arc $a_f$ occupies |
| Indexes and collections for speed allowance | |
| $\mathrm{S}_f$ | Speed level set for train $f$ |
| $\mathrm{s}_n$ | Speed level |
| $\mathrm{s}(a_f)$ | The associated speed level of the train-running arc $a_f$ |
| $\mathrm{W}_{a_f}$ | Speed allowance set that the train-running arc $a_f$ occupies |
| $\mathrm{w}_e^s(t)$ | Speed allowance |
| $\mathrm{S}_f^I$ | Interfering speed level set of the opposite direction for train $f$ |
| $\mathrm{s}^I(a_f)$ | Opposite direction interfering the speed level of train-running arc $a_f$ |
| $\mathrm{W}_{a_f}^I$ | Speed allowance of the opposite direction |
| Parameters | |
| $TW_{start}$ | The start time of the rolling time window |
| $TW_{end}$ | The end time of the rolling time window |
| $\tau$ | The rolling stepsize of the rolling time window |
| $F^{cand}$ | Candidate train set for the current rolling time window |
| $w_f$ | The weight of train $f$ |
| $c_{a_f}$ | The time duration indicated by the arc $a_f$ |
| Decision variables | |
| $x_{a_f}$ | Binary variable. 1 indicates that the arc $a_f$ is selected, and 0 otherwise |

### 3.2. Time-Space Network for Timetabling Description

We applied a time-space network to describe the out-of-gauge train rescheduling problem, as time-space networks can generalize complex train movements, and are therefore widely used to describe train traffic in the study of train timetabling (e.g., Caprara et al. [41], and Meng and Zhou [42]). The structure of the time-space network is shown in Figure 3.

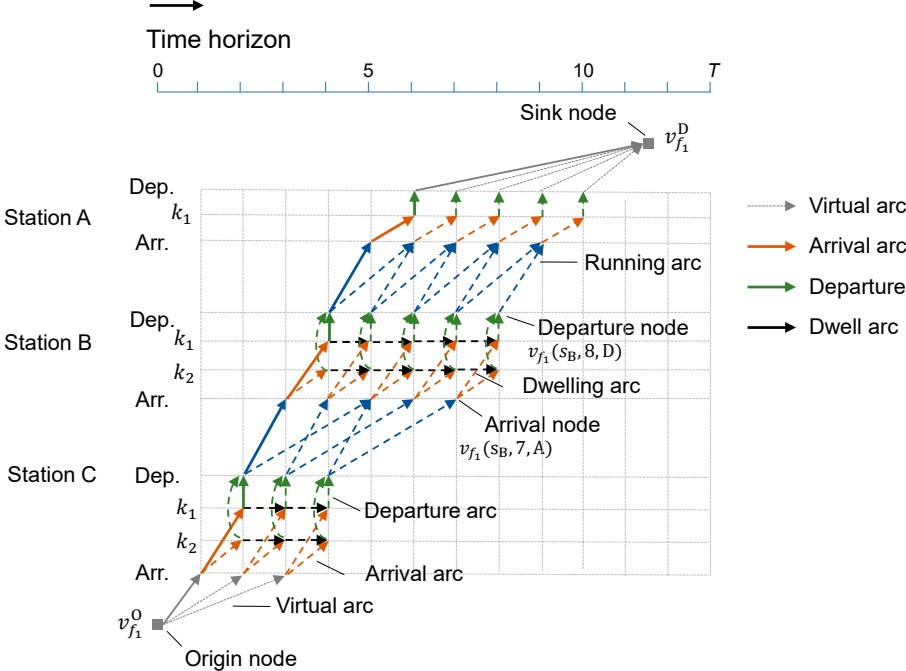

**Figure 3.** Time-space network considering speed variation and station track assignment.

A node of the time-space network is represented by an intersection of the grey grid in Figure 3. There are three types of nodes, as follows: Origin and sink nodes, which are denoted by $v_f^O$ and $v_f^S$, respectively, are the origin and sink point of the train path of train $f$. An arrival node $v_f(s, t, A)$ implies that train $f$ arrives at station $s$ at the moment $t$. A departure node $v_f(s, t, D)$ implies that train $f$ departs from station $s$ at the moment $t$. A track node $v_f(s, t, k)$ implies that train $f$ dwells at the track $k$ of station $s$ at the moment $t$.

An arc connecting two time-space nodes in the network represents a movement from one location to another, taking a particular duration. Correspondingly, the time-space network has five types of arcs, as follows: Virtual arcs are denoted by $a_f\left(v_f^O, v\right)$ for the origin virtual arc connecting the origin node $v_f^O$, and an arrival node $v$, or denoted by $a_f\left(v, v_f^S\right)$, for the sink virtual arc connecting a departure node $v$ and the sink node $v_f^S$. A train-running arc is denoted by $a_f(v, v')$, representing the train movement, where $v$ is a departure node, and $v'$ is an arrival node. In a segment, a train might run at different average speeds due to the speed reduction or the interference of the previous trains, resulting in many train-running arcs with different lengths from one departure node $v$. A train arrival arc is denoted by $a_f(v, v')$, representing the train moving from the border of a station to a certain station track, where $v$ is an arrival node and $v'$ is a track node. A train departure arc is denoted by $a_f(v, v')$, representing the train moving from a specific station track to the border of a station, where $v$ is a track node and $v'$ is a departure node. A train-dwelling arc is denoted by $a_f(v, v')$, representing the train dwelling, where $v$ is an arrival node and $v'$ is a departure node. Particularly, passing through a station is considered a special "dwelling" with a dwell time of 0.

The time-space-state network can be generated according to the timetabling parameters and scheduling rules of a single train. With a specified train path in the time-space

network, the train timetable can be interpreted through the time stamp of the arrival and departure nodes it traverses.

### 3.3. Time-Space Resources for Describing Train Conflicts

Note that the candidate trains do not share nodes and arcs in the time-space network. Instead, with the time-space network representation, we use the concept of the time-space resource (Liao et al. [23]) for describing the blocking section occupation in segments and platforms in stations (i.e., only at most can one train occupy the blocking section). We introduce two types of time-space resources for describing train conflicts on railway segments and platforms.

In the railway segment between two adjacent stations, a train consecutively occupies a series of blocking sections, as shown in Figure 4. Therefore, a train-running arc in the time-space network occupies a series of block sections for certain periods. For a train-running arc, the starting and ending time for occupying a block section can be calculated according to the performance parameter of the signal system. Therefore, the time-space resource occupied by arc $a_f$ can be built as $R_{a_f}$. The headway between two successive trains arriving at and departing from a station can be described explicitly by the occupation conflict of the time-space resources of the block sections. For example, in Figure 4, two trains, namely $f_1$ and $f_2$, traverse the section with three block sections. The train-running arcs of the two trains occupy a series of correspondent time-space resources (the occupying time-space resource set $R_{a_f}$ are denoted by corresponding color fills). With the restriction of occupation overlapping, the minimum departure and arrival headway remain six and five for these two trains at the segment.

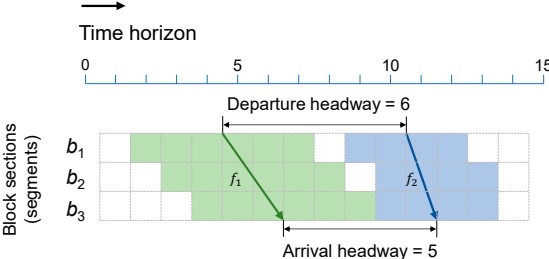

**Figure 4.** Segment block time-space resource.

Similarly, a train-dwelling arc occupies a series of time-space resources associated with the platform. The headway between two consecutive trains using the same platform can be described explicitly by the occupation conflict of the time-space resources of the platform. For example, in Figure 5, train $f_1$ and $f_2$ use the same platform at the station. Thus, with the restriction of occupation overlapping, the latter train, $f_2$, can enter the platform no earlier than the former train, $f_1$, leaving the platform, plus the safety time, as an interval.

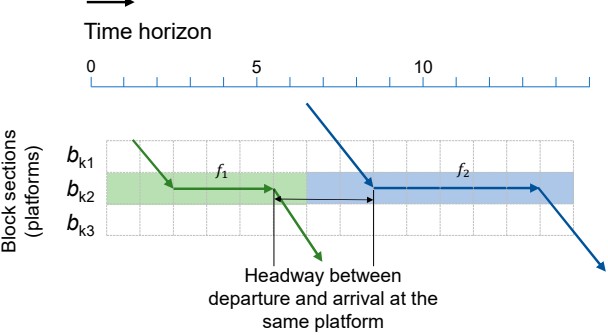

**Figure 5.** Platform time-space resource.

### 3.4. Speed Allowance for Denoting the Speed Restriction

In order to describe the speed restriction of trains due to the out-of-gauge train operation, we defined a speed level, set as $S_f = \left\{ s_1, s_2, \ldots, s_{|S_f|} \right\}$ for each train $f$. Each train-running arc $a_f$ has a corresponding speed level $s\left( a_f \right)$, as shown in Figure 6, where three train-running arcs, namely, $a_f^1$, $a_f^2$, and $a_f^3$, have a speed level of $s_1$, $s_2$, and $s_3$, respectively.

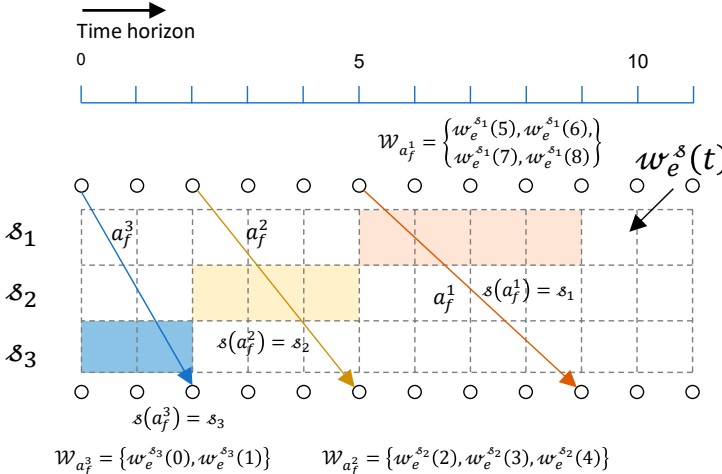

**Figure 6.** Train-running arc and its corresponding speed level.

To describe the speed restriction, we introduce a concept of speed allowance $w_e^s(t)$, as shown in Figure 6. Each speed allowance is displayed as a rectangular box in the matrix in Figure 6. If the train-running arc $a_f(v_1(s_1, t_1, D), v_2(s_2, t_2, A))$ is selected, the associated speed allowances in $W_{a_f} = \{ w_{e(s_1, s_2)}^{s(a_f)}(t) |, t(v_1) \leq t \leq t(v_2) \}$ are occupied.

If train $f$ is an out-of-gauge train, the running arc, $a_f$, might interfere with the trains in the opposite direction, resulting in the speed reduction of the opposite trains. Thus, we defined an interfering speed level, set in the opposite direction as $S_f^I = \left\{ s_1, s_2, \ldots, s_{|S_f|} \right\}$ for each out-of-gauge train $f$. Each out-of-gauge train-running arc, $a_f$, has a corresponding opposite direction interfering speed level $s^I\left( a_f \right)$, as shown in Figure 7. If the out-of-gauge train-running arc, $a_f$, is selected, the speed allowance of the opposite direction in $W_{a_f}^I = \{ w_{e(s_2, s_1)}^{s^I(a_f)}(t) |, t(v_1) \leq t \leq t(v_2) \}$ are interfered.

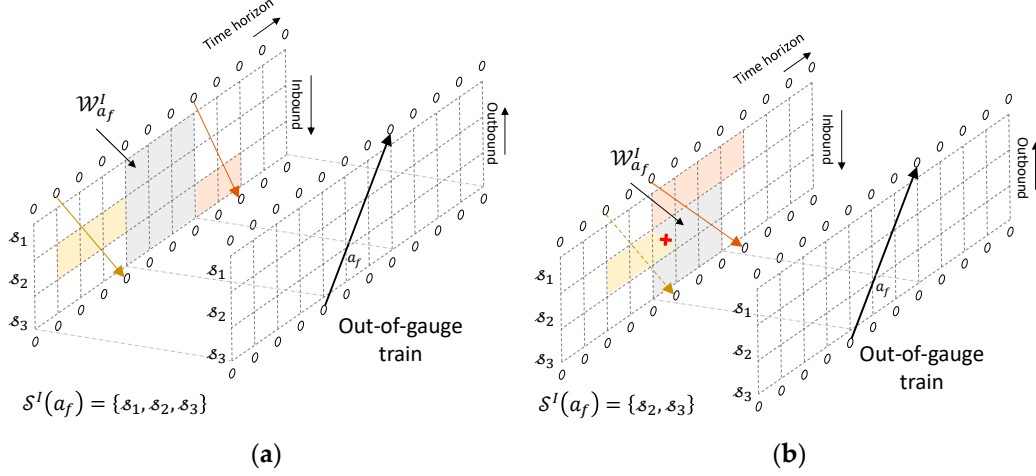

**Figure 7.** Out-of-gauge train-running arc and its interfering speed level of the opposite direction. (**a**) Temporal blockade of the opposite track; (**b**) Speed reduction of the opposite trains.

A speed allowance $w_e^s(t)$ can be occupied by a train-running arc, or interfered with by an out-of-gauge train-running arc in the opposite direction, or left spare. In other words, a speed allowance interfered with by an out-of-gauge train (in the opposite direction) can no longer be occupied by other trains. This occupation-interfering rule of speed allowance guarantees the speed restriction of the opposite-direction trains when an out-of-gauge train passes through the segment, as shown in Figure 8.

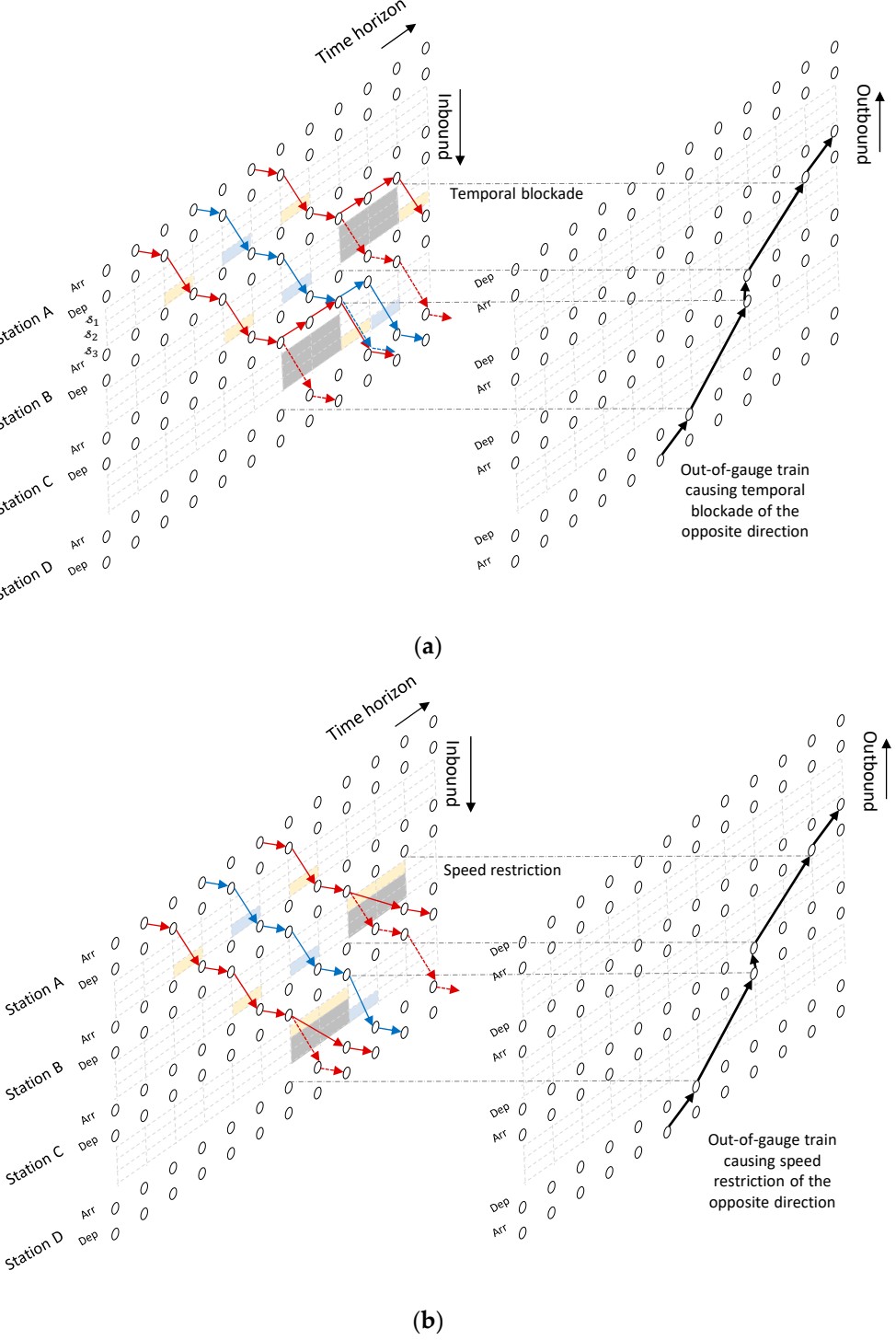

(**a**)

(**b**)

**Figure 8.** Train speed restrictions and temporal blockades considering the interfering of out-of-gauge trains. (**a**) temporal blockade; (**b**) speed restriction.

### 3.5. Optimization Model

For rescheduling the timetable considering the out-of-gauge trains, we built an integer programming model based on the time-space network proposed in Section 3.2 to minimize the total weighted delay of trains (i.e., the minimal cost time-space paths that are selected on the time-space network). The objective function of the train rescheduling problem can be written as follows.

$$\text{minimize} \sum_{f \in F} w_f \times \sum_{a_f \in A_f} c_{a_f} x_{a_f} \tag{1}$$

Subject to:

$$\sum_{a_f \in A_v^+} x_{a_f} = \sum_{a_f \in A_v^-} x_{a_f} \quad \forall f \in F, v \in V_f \tag{2}$$

$$\sum_{a_f \in A_{v_f^S}^+} x_{a_f} = \sum_{a_f \in A_{v_f^O}^-} x_{a_f} = 1 \quad \forall f \in F \tag{3}$$

$$\sum_{a_f : r \in R_{a_f}} x_{a_f} \leq 1 \quad \forall r \in R \tag{4}$$

$$x_{a_{f_1}} + x_{a_{f_2}} \leq 1$$
$$\forall w \in W, f_1, f_2 \in F, a_{f_1} \in A_{f_1} : w \in W_{a_{f_1}}, a_{f_2} \in A_{f_2} : w \in W_{a_{f_2}}^I \tag{5}$$

$$x_a \in \{0,1\} \quad \forall a \in A \tag{6}$$

The objective function (1) denotes the weighted total train delay compared with the original fundamental timetable. The train weight, $w_f$, is determined by the importance of the train. The impact of the train weights on the solutions would be further investigated in the case study. Constraints (2) and (3) are the flow balance constraints, which ensure that each scheduled train has a continuous time-space path from its origin node to its sink node. These constraints guarantee train-running continuity in both time and space dimensions. Constraint (4) is the generic headway constraint, denoting that, for the time-space resource, $r$, only at most, can one train-running arc occupy. This constraint ensures that no more than one train can occupy a time-space resource. If the time-space resource, $r$, is associated with a block section in a segment; this headway constraint denotes the arrival or departure headway. If the time-space resource, $r$, is associated with a station platform, this headway constraint denotes the headway between the departure and arrival time of two consecutive trains entering the same station platform. Constraint (5) is the speed allowance occupation constraint, which implicitly denotes the speed restriction caused by the out-of-gauge train running in the opposite direction. Based on the analysis in Section 3.4, for the speed allowance, w, when an out-of-gauge train-running arc, $a_{f_2}$, interferes with w, then the normal train-running arc, $a_{f_1}$, cannot occupy w. Therefore, the summation of the arc-selection binary variable, $x_{a_{f_1}}$ and $x_{a_{f_2}}$, must be less than or equal to one. Constraint (6) indicates the domain of the variables.

In the integer programming model, constraints (4) and (5) are the hard constraints of the model. Specifically, constraint (4) is a special "capacity" constraint, as the "capacity" of arcs is no longer an exclusive parameter of a single arc. Instead, the capacity is shared by multiple train service arcs through blocking resources denoted by constraint (4). Constraint (5) is the incompatible constraint of train-running arcs. By relaxing constraint (4) and (5), the problem can degrade into a series of min-cost flow problems for single trains. This nature of the model inspires us to apply appropriate heuristic algorithms to solve the model efficiently.

### 3.6. Rolling Time Horizon Solution Method

The model is an integer programming model, which can be solved by commercial solvers. However, due to the complexity of the train-rescheduling problem, the size of the proposed time-space network is very large when modeling real world cases. As the

very large-scale instances are difficult to be solved by commercial solvers (the detailed comparison and analysis are introduced in Section 4.2), for the large-scale instances, a rolling-time horizon approach can be applied to deconstruct the entire problem into several time-dependent sub-problems that can be solved consecutively, as shown in Figure 9. We only include trains that possibly originate at this time window for each rolling-time window and neglect the other trains. When the solution of a rolling-time window is obtained, we mark the occupation of blocking resources and the occupancy and interference of the speed allowance. These marks are delivered with the time window rolling to the next position. Therefore, the occupation of blocking resources, and the occupation and interference of speed allowance in previous time windows, are considered known constraints while scheduling the trains in the next time windows.

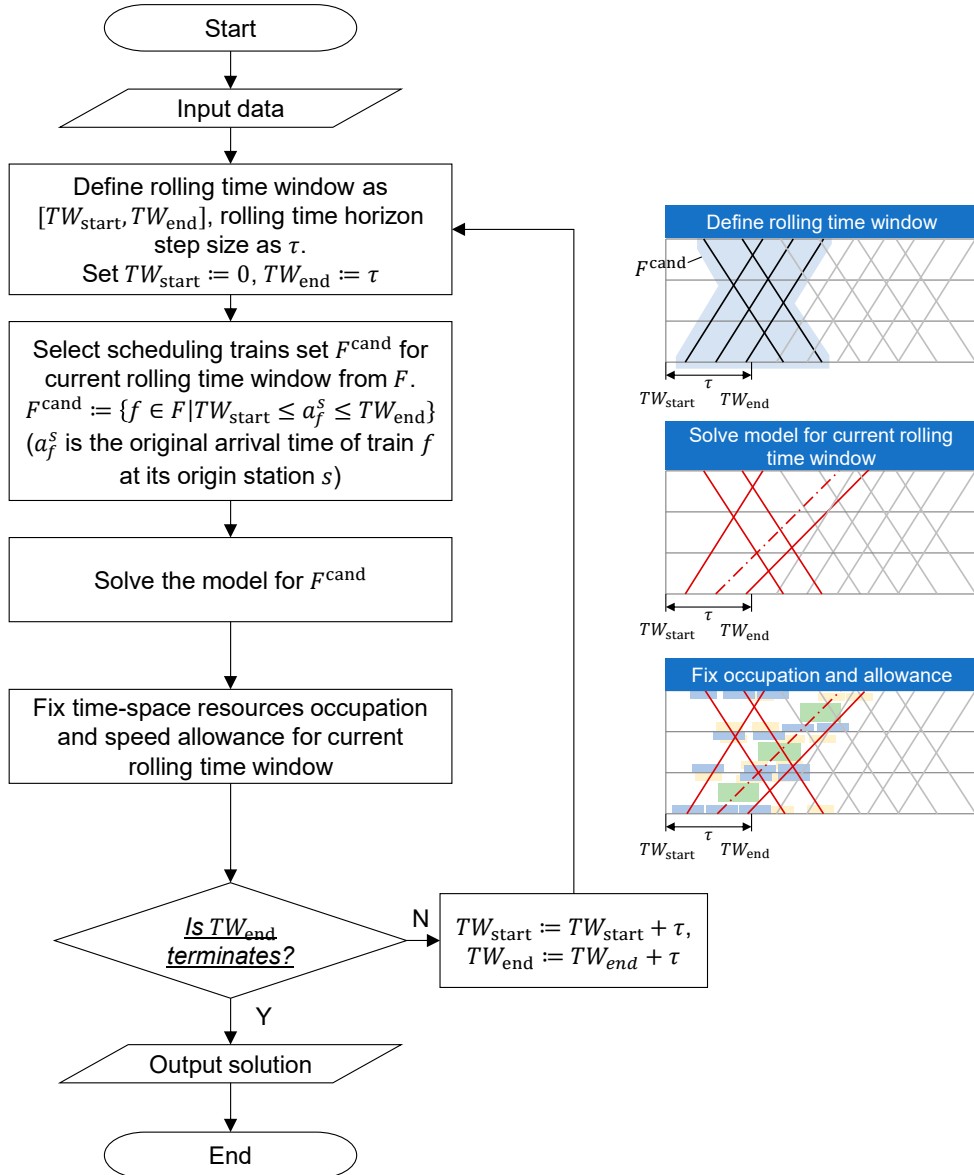

**Figure 9.** Rolling-time horizon approach for solving the out-of-gauge train-rescheduling problem.

Considering the high priority of passenger trains, we conduct two rounds of a schedule-and-fix process by executing, then totalizing, the rolling-time horizon approach. The first round is for passenger trains, where the passenger trains are scheduled without considering the freight trains, while the second round is to schedule freight trains with a derived and fixed-passenger train timetable. This two-round rolling schedule approach guarantees that

high-priority trains will not be substantially interfered with by low-priority trains (the out-of-gauge trains, in particular). The benefit of the two-round rolling schedule approach will also be demonstrated in Section 4.2.

A simplified illustrative pseudocode of the rolling-time horizon algorithm is shown as follows Algorithm 1.

---

**Algorithm 1: The rolling time horizon procedure**

| | |
|---|---|
| 1: | Load infrastructure and train data |
| 2: | **Foreach** r ∈ R (defining the rounds of rolling time horizon) |
| 3: | Generate train set $F_r$ according to train type |
| 4: | Set initial parameters $TW_{start} := 0$, $TW_{end} := \tau$ |
| 5: | **While** $TW_{end} \leq T$: |
| 6: | Generate candidate train set $F^{cand} := \{f \in F_r | TW_{start} \leq a^s_f \leq TW_{end}\}$ |
| 7: | Use Gurobi to solve model (1) to (6) for $F = F^{cand}$, obtain selected arc set $A^{sel}$ |
| 8: | Fix resource occupation and speed allowance according to $A^{sel}$ |
| 9: | $TW_{start} := TW_{start} + \tau$; $TW_{end} := TW_{end} + \tau$ |
| 10: | **End While** |
| 11: | **End For** |
| 12: | Output solution and terminate the algorithm |

---

## 4. Results and Discussions

### 4.1. Experiment Setup

The dataset applied in the experiment comes from one of the busiest conventional railway lines in China with mixed-passenger and freight train traffic. The railway line section consists of ten stations and nine double-track segments. There are 23 passenger trains running between ZZ and XC station, and 21 freight trains running between ZZN and XC station from 17:00 to 22:00. The number of side tracks for each station is three, allowing overtaking. The running time of passenger and freight trains can be referred to in Appendix A. In the case study, we apply two series of timetables to demonstrate the computational performance of the rescheduling methods. One timetable is artificially generated. The timetable which is produced, with the provided parameters and regulations, is applied to demonstrate the computational performance and the relations between the timetabling parameters and timetable performance. The other timetable is a practical timetable of a conventional railway line, as shown in Figure 10. The weight of passenger trains is set to 10, while the non- and out-of-gauge freight trains are set to 1 and 0.4. We consider two freight trains (namely F5 and F10) as out-of-gauge trains and reschedule the timetable with the out-of-gauge trains, and analyze the timetable patterns of out-of-gauge trains in detail. The out-of-gauge train in the experiment can be classified into two types, namely, Level-1 and Level-2.

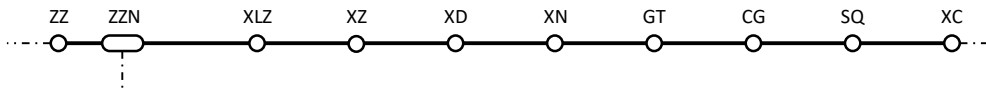

**Figure 10.** The conventional railway line of which the infrastructure data is used in the case study.

We apply the following temporal blockade and speed restriction rules in the experiment:

- Level-1 out-of-gauge trains can run at 60 km/h with 140 km/h speed restriction on the parallel track, or run at 90 km/h with the temporal blockade on the parallel track.
- Level-2 out-of-gauge trains can only run at 60 km/h with the temporal blockade on the parallel track.

In the rolling-time horizon approach, for each rolling-time window, we solve the optimization model by the Gurobi 9.5.0 solver. The rolling-time horizon framework is programmed in C#, which runs on the .Net 6.0 runtime. The program for this experiment runs on a PC with AMD R9 5900 CPU and 64 GB internal memory.

We apply an artificially generated train dataset to demonstrate the computational performance of the proposed algorithm, compared with the benchmark solution approaches. For reporting the solution of different instances, we name the instance using the "rolling-time horizon length", "total number of trains", "number of the out-of-gauge train", and "out-of-gauge level" format. The convergence analysis and the computational quality and efficiency of the models, with different constraints and input data, are provided. Meanwhile, we also apply a real-world dataset with realistic numbers and combinations of passenger and freight trains, to experiment with rescheduling out-of-gauge trains in a practical rescheduling task.

*4.2. Benchmark Solution Comparison*

Based on the time-space network-based integer programming model, we set up a series of computational performance experiments, using the commercial solver Gurobi, to report solution efficiency in terms of different sizes of the model. Specifically, we calculate the artificially generated cases with the entire planning-time horizons varying from 60 to 180 min, containing different amounts of trains that need to be rescheduled. Moreover, with the same number of trains, we change the proportion of out-of-gauge trains with different levels to investigate if the number and the level of out-of-gauge trains impact the solution difficulties of the model. Figure 11 reports the convergence of the optimality gaps with the computational time, by the proportion of out-of-gauge trains set to 60, 120, and 180 min planning-time horizons.

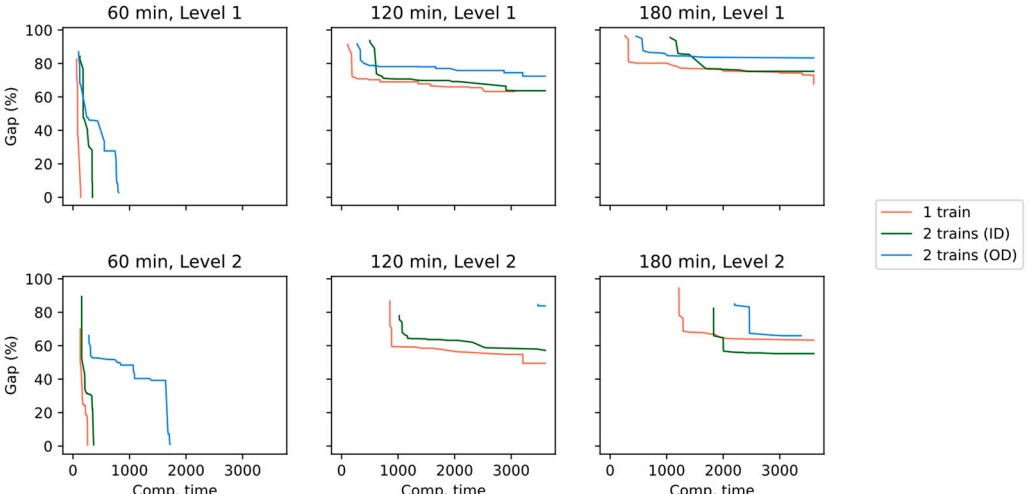

**Figure 11.** Convergence of the optimality gap (ID for identical direction; OD of opposite direction).

From Figure 11, we can conclude that the optimization horizon length (i.e., the total number of trains) significantly impacts the computational time. When the time horizon is 60 min, the solver can obtain the optimal solutions within 1800 s of computational time, regardless of the number and the level of out-of-gauge trains. However, when the optimization horizon length reaches 120 min, the solver cannot obtain optimal solutions for all instances within 3600 s. Even still, the computational time positively correlates with the number of out-of-gauge trains. Under the same conditions, the cases with two out-of-gauge trains always run longer than the case with one out-of-gauge train, and the cases with two opposite directions out-of-gauge trains spend a longer computational time than the cases with two identical directional out-of-gauge trains. The level of out-of-gauge trains significantly impacts the computational time for finding the first feasible solution, but has less impact on the optimality gap by 3600 s. This is because the level one out-of-gauge trains have two-speed level options, which implies that the solutions have great flexibility. However, the super out-of-gauge trains result in the blockade of the parallel track, which might interfere greatly with the traffic of the parallel track.

From the analysis above, we can draw the following conclusions:

1.  Considering the speed reduction caused by out-of-gauge trains further increases the solution complexity of the train-rescheduling problem, which is already recognized as a very combinatorial problem. Thus, the decomposition heuristic, such as a rolling-time horizon approach, is very necessary to be applied to solve the problems on a practical scale.

2.  The computational efficiency of the problem depends not only on the number of stations and trains but also depends on the number (or proportion) of the out-of-gauge trains, as well as the level of out-of-gauge trains, as different levels of out-of-gauge trains results in various impacts on the original railway traffic.

3.  We further compare the solution quality and computational efficiency of several benchmark solution approaches. Among the solution methods, Gurobi is a popular integer-programming solver that has extraordinary computational performance for solving large-scale instances, and is therefore commonly chosen as the computational quality benchmark. Besides the commercial solver Gurobi, we apply a series of rolling-time horizon ("RH" for short) approaches. The RH-solution method is a one-round rolling-time horizon approach without classifying the train types during the rolling horizon solution procedure. As the passenger trains usually have higher priority when rescheduling the out-of-gauge trains, we consider scheduling the passenger trains without freight trains, and rescheduling the freight trains (including normal and out-of-gauge trains together), namely, a two-round P-RH solution method. This approach hopefully improves the solution quality, especially when the passenger trains have relatively high priority. Lastly, we include another two-round solution method named OOG-RH. In this method, the passenger trains and the non-out-of-gauge trains are scheduled in the first round of the rolling time window approach, and the out-of-gauge trains are then scheduled with an obtained timetable derived from the first round. The OOG-RH solution method guarantees the punctuality of the passenger and non-out-of-gauge trains to the most extent. The objective value and the computational time are displayed in Figures 12 and 13. The detail train delay of different train categories, the number of extra stops, and the number of speed reduction applied is reported in Tables 4 and 5.

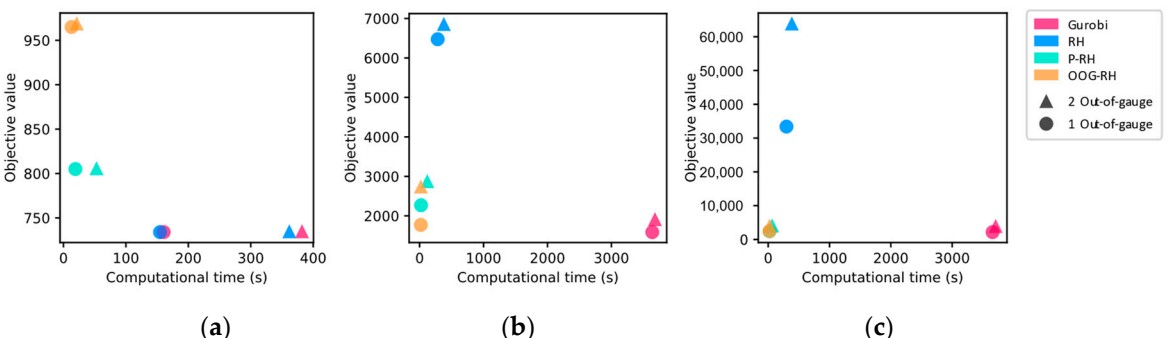

(**a**)    (**b**)    (**c**)

**Figure 12.** Computational performance of different solution methods for level-1 out-of-gauge trains. (**a**) 60 min planning-time horizon; (**b**) 120 min planning-time horizon; (**c**) 180 min planning-time horizon.

We also display the computational time and the corresponding objective value of the artificial generated instances in Figures 12 and 13.

From Figures 12 and 13, and Tables 4 and 5, we can conclude that the three rolling horizon approaches usually calculate much faster than Gurobi, except for the very small instances. However, the solution qualities are usually worse than the Gurobi approaches. Among the rolling horizon cases, the simple rolling horizon approach performs well in the 60 min and 120 min cases, but performs substantially worse in large-scale cases. The passenger train priority rolling horizon approach results in significant freight train delays (both the number and the total delay time), compared with the rescheduled out-of-gauge train-rolling horizon approach. Overall, in case of the passenger train delay caused by the

out-of-gauge trains, we recommend that the passenger train should be fixed and remain unchanged while rescheduling the out-of-gauge trains.

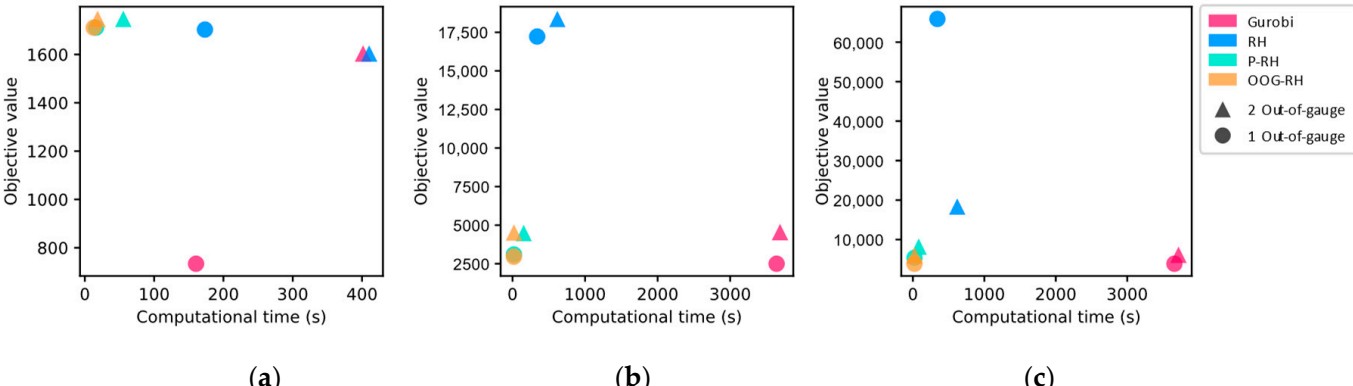

(**a**)                (**b**)                (**c**)

**Figure 13.** Computational performance of different solution methods for level-2 out-of-gauge trains. (**a**) 60 min planning-time horizon; (**b**) 120 min planning-time horizon; (**c**) 180 min planning-time horizon.

**Table 4.** Quality and efficiency comparisons between benchmarks solution methods (out-of-gauge train Level-1).

| Instance | Solution Method * | # Delay Trains | | | Delay Time (min) | | | # Interfered Speed Reduction | # Blockade |
|---|---|---|---|---|---|---|---|---|---|
| | | Pas. | Frei. | Out-of-Gauge | Pas. | Frei. | Out-of-Gauge | | |
| 60-12-1-Level-1 | Gurobi | 0 | 1 | 1 | 0 | 734 | 679 | 12 | 6 |
| | RH | 0 | 1 | 1 | 0 | 734 | 679 | 12 | 6 |
| | P-RH | 0 | 5 | 1 | 0 | 805 | 508 | 10 | 6 |
| | OOG-RH | 0 | 1 | 1 | 0 | 965 | 910 | 8 | 6 |
| 60-12-2-Level-1 | Gurobi | 0 | 1 | 1 | 0 | 735 | 735 | 11 | 9 |
| | RH | 0 | 1 | 1 | 0 | 735 | 735 | 11 | 9 |
| | P-RH | 0 | 5 | 1 | 0 | 806 | 564 | 11 | 9 |
| | OOG-RH | 0 | 1 | 1 | 0 | 969 | 969 | 10 | 8 |
| 120-21-1-Level-1 | Gurobi | 1 | 6 | 1 | 12 | 1348 | 1120 | 14 | 5 |
| | RH | 1 | 10 | 1 | 237 | 1735 | 609 | 26 | 5 |
| | P-RH | 0 | 9 | 1 | 0 | 2270 | 607 | 26 | 4 |
| | OOG-RH | 0 | 1 | 1 | 0 | 1771 | 1771 | 12 | 5 |
| 120-21-2-Level-1 | Gurobi | 0 | 12 | 2 | 0 | 1918 | 817 | 24 | 9 |
| | RH | 1 | 10 | 2 | 235 | 2161 | 986 | 30 | 9 |
| | P-RH | 0 | 10 | 2 | 0 | 2882 | 1003 | 30 | 7 |
| | OOG-RH | 0 | 2 | 2 | 0 | 2745 | 2745 | 12 | 12 |
| 180-29-1-Level-1 | Gurobi | 2 | 5 | 1 | 23 | 1713 | 760 | 29 | 2 |
| | RH | 6 | 11 | 1 | 1573 | 1945 | 609 | 48 | 5 |
| | P-RH | 0 | 9 | 1 | 1 | 2426 | 628 | 32 | 1 |
| | OOG-RH | 0 | 1 | 1 | 1 | 2428 | 2428 | 16 | 4 |
| 180-29-2-Level-1 | Gurobi | 0 | 10 | 2 | 1 | 3970 | 3210 | 31 | 10 |
| | RH | 8 | 11 | 2 | 3074 | 2449 | 986 | 54 | 9 |
| | P-RH | 0 | 11 | 2 | 1 | 4138 | 2056 | 37 | 5 |
| | OOG-RH | 0 | 2 | 2 | 1 | 4028 | 4028 | 19 | 8 |

* Gurobi for directly solved by Gurobi solver; RH for rolling horizon approach combining passenger and freight trains together; P-RH for two-round rolling-time horizon approach, where passenger trains are rescheduled in the first round; OOG-RH for two-round rolling horizon approach, where out-of-gauge trains are rescheduled in the last round.

**Table 5.** Quality and efficiency comparisons between benchmarks solution methods (out-of-gauge train Level-2).

| Instance | Solution Method * | # Delay Trains | | | Delay Time (min) | | | # Interfered Speed Reduction | # Blockade |
|---|---|---|---|---|---|---|---|---|---|
| | | Pas. | Frei. | Out-of-Gauge | Pas. | Frei. | Out-of-Gauge | | |
| 60-12-1-Level-2 | Gurobi | 0 | 1 | 1 | 0 | 734 | 679 | 12 | 6 |
| | RH | 0 | 2 | 1 | 0 | 938 | 327 | 9 | 8 |
| | P-RH | 0 | 1 | 1 | 0 | 1711 | 1656 | 6 | 8 |
| | OOG-RH | 0 | 1 | 1 | 0 | 1711 | 1656 | 6 | 8 |
| 60-12-2-Level-2 | Gurobi | 1 | 5 | 1 | 4 | 1524 | 1331 | 6 | 12 |
| | RH | 1 | 5 | 1 | 4 | 1524 | 1331 | 6 | 12 |
| | P-RH | 0 | 1 | 1 | 0 | 1747 | 1747 | 6 | 12 |
| | OOG-RH | 0 | 1 | 1 | 0 | 1747 | 1747 | 6 | 12 |
| 120-21-1-Level-2 | Gurobi | 0 | 2 | 1 | 0 | 2501 | 2470 | 10 | 8 |
| | RH | 2 | 4 | 1 | 725 | 2724 | 1537 | 24 | 8 |
| | P-RH | 0 | 4 | 1 | 0 | 3103 | 1820 | 25 | 8 |
| | OOG-RH | 0 | 1 | 1 | 0 | 2960 | 2960 | 9 | 8 |
| 120-21-2-Level-2 | Gurobi | 1 | 5 | 2 | 4 | 4472 | 4252 | 12 | 16 |
| | RH | 2 | 7 | 2 | 725 | 3866 | 2688 | 24 | 16 |
| | P-RH | 0 | 7 | 2 | 0 | 4488 | 3105 | 25 | 16 |
| | OOG-RH | 0 | 2 | 2 | 0 | 4524 | 4524 | 8 | 16 |
| 180-29-1-Level-2 | Gurobi | 0 | 1 | 1 | 1 | 3866 | 3866 | 15 | 8 |
| | RH | 9 | 5 | 1 | 3142 | 3039 | 1537 | 56 | 8 |
| | P-RH | 0 | 7 | 1 | 1 | 5357 | 2614 | 20 | 8 |
| | OOG-RH | 0 | 1 | 1 | 1 | 3866 | 3866 | 12 | 8 |
| 180-29-2-Level-2 | Gurobi | 0 | 2 | 2 | 1 | 6137 | 6137 | 18 | 16 |
| | RH | 2 | 7 | 2 | 725 | 3866 | 2688 | 24 | 16 |
| | P-RH | 0 | 8 | 2 | 1 | 8195 | 5195 | 19 | 16 |
| | OOG-RH | 0 | 2 | 2 | 1 | 6016 | 6016 | 11 | 16 |

* Gurobi for directly solved by Gurobi solver; RH for rolling horizon approach combining passenger and freight trains together; P-RH for two-round rolling-time horizon approach, where passenger trains are rescheduled in the first round; OOG-RH for two-round rolling horizon approach, where out-of-gauge trains are rescheduled in the last round.

### 4.3. Performance on Practical Timetables

In the practical train rescheduling cases, we assume that the scheduling time horizon is (17:00 to 23:00). During this period, passenger and freight trains are run with a high level of mixed-railway traffic. This period is a busy period for operating passenger trains. As the intensity of the train paths is substantial, the out-of-gauge train has a very significant impact on the passenger and non-out-of-gauge freight trains. Rescheduling such an intensive timetable, considering the temporal blockades and speed restrictions, would prove very challenging. There are two freight trains labeled as out-of-gauge trains and, the train paths passing through the segment later than 17:00, must be rescheduled. One out-of-gauge train (No. F5) running from ZZN to XC is an out-of-gauge Level-1. This train can run with a technical speed of 90 km/h, with the blockade of the parallel track, or optionally run with a technical speed of 60 km/h, with a 140 km/h speed restriction of the parallel track. The other out-of-gauge train (No. F10) running from XC to ZZN is a Level-2 out-of-gauge. This train can only run with a technical speed of 60 km/h with the blockade of the parallel track.

In this scheduling-time horizon, there are 16 passenger trains and 7 freight trains downward, and 7 passenger trains and 14 freight trains upward. The passenger trains run with technical speeds between 140 km/h to 160 km/h, while the freight trains run with technical speeds between 90 km/h to 120 km/h. The original train timetable is shown in Figure 14.

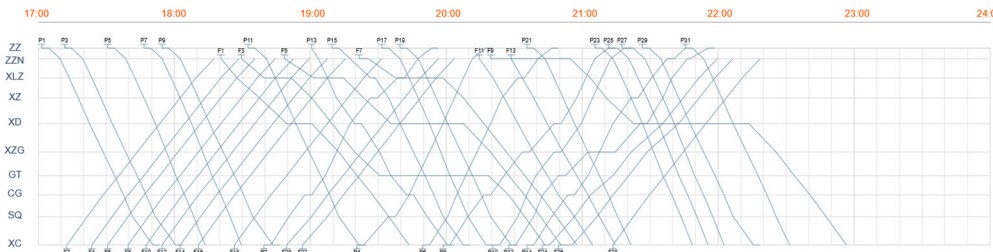

**Figure 14.** The original train timetable for the practical case study.

The rolling-time horizon approach is a simple decomposition technique that uses the output of the last rolling-time window as the input of the following rolling-time window. Although the solutions of each rolling-time window can be solved to be optimal, the final, entire, timetable solution still cannot guarantee optimality due to a lack of communication among different rolling-time windows. Nevertheless, the solution quality of the rolling-time horizon approach can be improved by setting the appropriate rolling-time horizon lengths and the rolling step-sizes. Therefore, we perform a series of case studies to compare the solution quality and efficiency. The optimality gaps, objective values, and the computational time of different rolling-time horizon lengths, are shown in Table 6 and Figure 15.

From Table 6 and Figure 15, we can conclude that the cases with longer rolling horizon length typically have better timetable quality, but consume a longer computational time. The case with a full-length rolling-time horizon (directly solved by the commercial solver without a rolling-time horizon approach) reaches the objective value of 26,554.2 min, using 3757 s of computational time. However, the case with 10 min rolling-time horizon length has the worst quality. As the train departure headway is 6 min, using 10 min rolling-time horizon is almost equivalent to train-by-train scheduling. However, the shorter rolling-time horizon length does not always result in a shorter computational time, as the setup of the commercial solver is time consuming; the frequent solver setup might reduce the efficiency of the calculation.

**Table 6.** Computational time by the step-size and rolling-time window.

| Rolling Time Window Length (min) | Obj. Value (min) | Computational Time (s) | Passenger Delay Time (min) | Freight Train Delay Time (min) | Out-of-Gauge Train Delay Time (min) |
|---|---|---|---|---|---|
| (Full length, 360 min) | 26,554.2 | 3757 | 82 | 956 | 390 |
| 10 | 7416.8 | 46 | 2 | 735 | 289 |
| 20 | 7416.8 | 44 | 2 | 735 | 289 |
| 30 | 7416.8 | 40 | 2 | 735 | 289 |
| 40 | 7416.8 | 45 | 2 | 735 | 289 |
| 50 | 7480.8 | 40 | 2 | 733 | 314 |
| 60 | 7416.8 | 53 | 2 | 735 | 289 |
| 70 | 7416.8 | 48 | 2 | 735 | 289 |
| 80 | 7480.8 | 49 | 2 | 733 | 314 |
| 90 | 7480.8 | 51 | 2 | 733 | 314 |
| 100 | 7043.6 | 157 | 2 | 729 | 384 |
| 110 | 7414.8 | 95 | 2 | 735 | 289 |
| 120 | 6524 | 113 | 2 | 745 | 314 |
| 130 | 6524 | 113 | 2 | 745 | 314 |
| 140 | 6524 | 113 | 2 | 745 | 314 |
| 150 | 5368.6 | 122 | 2 | 607 | 314 |
| 160 | 5368.6 | 122 | 2 | 607 | 314 |
| 170 | 5368.6 | 119 | 2 | 607 | 314 |
| 180 | 5368.6 | 124 | 2 | 607 | 314 |

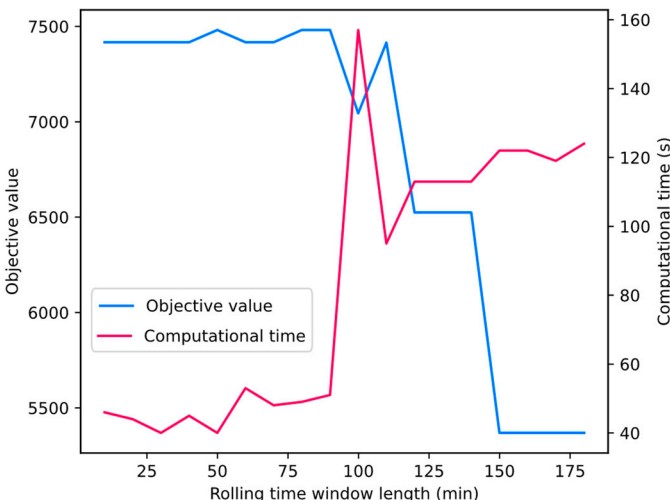

**Figure 15.** The computational performances by rolling-time horizon length.

The solution timetable derived from the algorithm, with a horizon length of 300 min, is shown in Figure 16. The passenger trains are granted "P", and freight trains are granted "F". The train paths of out-of-gauge trains are displayed with stronger lines. The color of the train paths shows the train delays compared with the original timetable (delays less than 5 min, delays between 5 to 30 min, and delays over 30 min are marked with blue, yellow, and red, respectively).

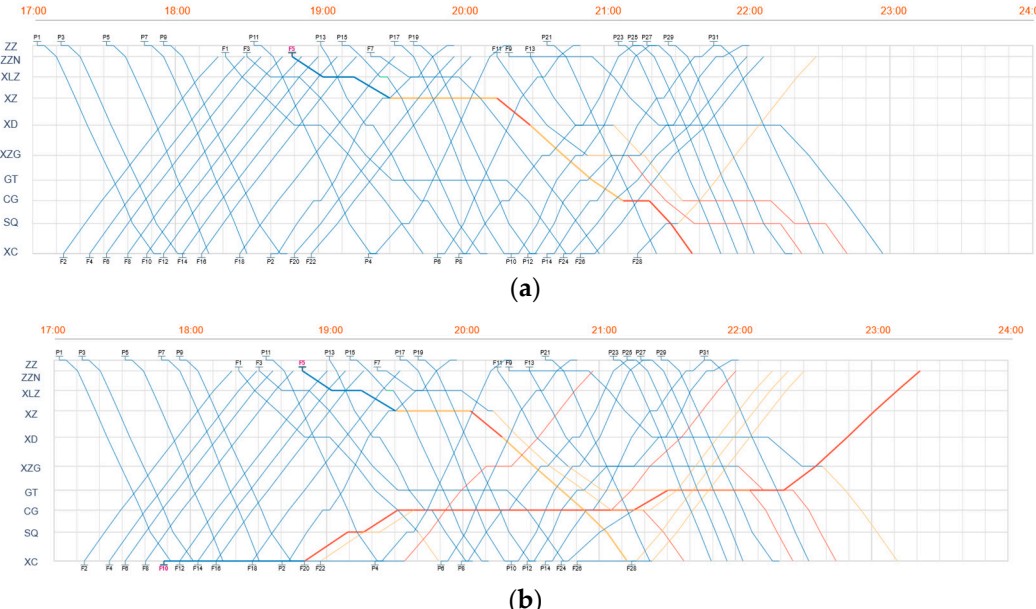

**Figure 16.** Solution timetable with one and two out-of-gauge trains. (**a**) One out-of-gauge train; (**b**) Two out-of-gauge trains.

In Figure 16a, train F5 is a Level-1 out-of-gauge train. It runs at 60 km/h between ZZN and CG to avoid the speed reduction of the opposite direction trains. However, thanks to the extra dwell-time arranged by train F28, train F5 can run at 90 km/h between CG and XC, as no opposite direction train is scheduled to meet F5 in these segments. F11 and F13 have consecutive delays caused by the out-of-gauge train F5. In Figure 16b, besides the Level-1 out-of-gauge train F5, train F10 is a Level-2 out-of-gauge train, which cannot meet any trains along its path. Due to the severe impact on the opposite direction, the Level-2 out-of-gauge train, F10, tends to use the space after 22:00 when no opposite direction trains are scheduled in the original timetable, to avoid interference on both the opposite direction

and the identical direction. The rescheduled train path of F5 and F10 results in four identical direction train delays (222 min in total) and six opposite direction train delays (342 min in total).

In the practical cases, we also compare the solutions of the following four strategies. One solution approach is to solve by Gurobi directly; the rest of the three are rolling-time horizon approaches. P-RH is a two-round rolling-time horizon approach where the first round is passenger train rescheduling, and the second round is freight train rescheduling. OOG-RH is a two-round rolling-time horizon approach where the first round is the combination of passenger and non-out-of-gauge train rescheduling, and the second round is out-of-gauge train scheduling, respectively. RH is a one-round rolling-time horizon approach where all trains are scheduled together. The performance indexes are reported in Table 7.

**Table 7.** Performance indexes of different rolling-time strategies.

| Strategy * | Feasibility | Obj Value | Passenger Train Delay (min) | Freight Train Delay (min) | Out-of-Gauge Train Delay (min) | # Speed Reduction | # Segment Blockage |
|---|---|---|---|---|---|---|---|
| P-RH | Feasible | 5368.6 | 7 | 6963 | 4336 | 38 | 10 |
| OOG-RH | Feasible | 4498.4 | 7 | 7264 | 7264 | 35 | 13 |
| RH | Infeasible (18 train failed) | 2032.2 | 1 | 3333 | 3302 | 12 | 13 |

* RH for rolling horizon approach combining passenger and freight trains together; P-RH for two-round rolling-time horizon approach where passenger trains are rescheduled in the first round; OOG-RH for two-round rolling horizon approach where out-of-gauge trains are rescheduled in the last round.

Table 7 shows that, in general, the commercial solver has worse performance in large-scale cases. The P-RH strategy has a significantly larger number of delayed freight trains compared with OOG-RH. Since the weight of non-out-of-gauge freight trains is larger than out-of-gauge trains, the objective value of P-RH is greater than OOG-RH. Among the rolling-time horizon strategies, the OOG-RH strategy can obtain the best objective value. However, the prior scheduling of the non-out-of-gauge trains occupies the spaces on the timetable without considering the out-of-freight trains and results in the extremely long delay of the out-of-gauge trains, so as to lengthen the total delay of freight trains. RH strategy cannot obtain any feasible solution during the solution procedure. Therefore, the rolling-time horizon strategies should be in accordance with the weight of trains and the number of out-of-gauge trains.

## 5. Conclusions

This paper addresses a train rescheduling problem considering the speed reduction and temporal blockade, due to the special dispatching rules of out-of-gauge trains. We first analyze the impact of the out-of-gauge trains on other trains running in identical and opposite directions. In order to model this impact, a novel concept of speed allowance is proposed to describe the interaction between the out-of-gauge train and the trains running in the opposite direction. A time-space network, including variable train-running time and station track assignment, is applied to model the train dynamics, followed by an integer programming model and a rolling-time horizon solution approach.

The case study shows that, for large-scale instances, the time-space network-based integer-programming model is difficult to solve, and the rolling-time horizon approach is necessary to accelerate the solution procedure. The rolling-time horizon strategy selection is essential for the solution's quality and efficiency. This method is useful for the railway dispatcher to build the daily timetable when out-of-gauge trains run efficiently. The foreseeable delay of passenger and freight trains due to the out-of-gauge trains can be notified to the associated staff and the public in time for the organization. The Gurobi solver cannot obtain an optimal solution within 1 h when the planning-time horizon is greater than 120 min. The two-round rolling-time window strategy OOG-RH has the best computational performance for large-scale instances. With the rolling-time horizon

approach, the rescheduled timetable can be obtained within 124 s for the 300 min planning-time horizon using a 180 min rolling-time window. The width of the rolling-time windows should be reasonably determined for the trade-off between solution quality and efficiency.

The limitations of this research include computational quality and efficiency. The proposed rolling horizon approach is a temporal-based decomposition method. When the length of the railway line is extremely long, it would be very difficult to obtain a good quality solution even in a rolling-time window, which limits the geographical scale of the method applied. The rolling-time horizon approach is without feedback from the latter rolling-time windows to earlier ones. The lack of backward feedback might lead to the local optimum, especially for the late rolling-time window. Even still, the train running time used in the train rescheduling can be calculated more concretely, as the opposite-direction train can recover the normal speed right after the meeting point (instead of exiting the segment) with the out-of-gauge train.

For future research, we may manage to improve the solution quality and efficiency by applying more elaborated algorithms, such as the metaheuristic-combining exact method, Lagrangian relaxation, or column generation. Additionally, the impact of the out-of-gauge train schedule on railway capacity, as well as the trade-off between the passenger and freight trains, can be thoroughly investigated in the future in a more sophisticated experiment.

**Author Contributions:** Conceptualization, methodology, formal analysis, writing—original draft preparation, writing—review and editing Z.L. The author has read and agreed to the published version of the manuscript.

**Funding:** This work is supported by the Science and Technology Research and Development Project of China State Railway Group Company, Ltd. (Grant No. N2022X018).

**Data Availability Statement:** The data applied in the research is unavailable due to privacy.

**Acknowledgments:** The author sincerely thanks the technical support from Lingyun Meng, Jianrui Miao, and Xiaojie Luan.

**Conflicts of Interest:** The authors declare no conflict of interest.

## Appendix A

The running time applied in the experiment is as Table A1.

**Table A1.** Train running times.

| Segment (Directed) | Speed Limitation (km/h) | Free Flow Running Time (min) | Additional Running Time (min) | |
|---|---|---|---|---|
| | | | Acceleration | Deceleration |
| ZZ-ZZN | 140 | 3 | 2 | 1 |
| ZZN-XLZ | 140 | 4 | 2 | 1 |
| XLZ-XZ | 140 | 4 | 2 | 1 |
| XZ-XD | 140 | 5 | 2 | 1 |
| XD-XZG | 140 | 6 | 2 | 1 |
| XZG-GT | 140 | 5 | 2 | 1 |
| GT-CG | 140 | 4 | 2 | 1 |
| CG-SQ | 140 | 4 | 2 | 1 |
| SQ-XC | 140 | 6 | 2 | 1 |
| ZZN-ZZ | 140 | 3 | 2 | 1 |
| XLZ-ZZN | 140 | 4 | 2 | 1 |
| XZ-XLZ | 140 | 4 | 2 | 1 |
| XD-XZ | 140 | 5 | 2 | 1 |
| XZG-XD | 140 | 6 | 2 | 1 |
| GT-XZG | 140 | 5 | 2 | 1 |
| CG-GT | 140 | 4 | 2 | 1 |
| SQ-CG | 140 | 4 | 2 | 1 |
| XC-SQ | 140 | 6 | 2 | 1 |
| ZZ-ZZN | 90 | 4 | 2 | 3 |

**Table A1.** *Cont.*

| Segment (Directed) | Speed Limitation (km/h) | Free Flow Running Time (min) | Additional Running Time (min) | |
|---|---|---|---|---|
| | | | Acceleration | Deceleration |
| ZZN-XLZ | 90 | 7 | 2 | 3 |
| XLZ-XZ | 90 | 7 | 2 | 3 |
| XZ-XD | 90 | 9 | 2 | 3 |
| XD-XZG | 90 | 10 | 2 | 3 |
| XZG-GT | 90 | 8 | 2 | 3 |
| GT-CG | 90 | 7 | 2 | 3 |
| CG-SQ | 90 | 7 | 2 | 3 |
| SQ-XC | 90 | 9 | 2 | 3 |
| ZZN-ZZ | 90 | 5 | 2 | 3 |
| XLZ-ZZN | 90 | 8 | 2 | 3 |
| XZ-XLZ | 90 | 7 | 2 | 3 |
| XD-XZ | 90 | 9 | 2 | 3 |
| XZG-XD | 90 | 10 | 2 | 3 |
| GT-XZG | 90 | 8 | 2 | 3 |
| CG-GT | 90 | 7 | 2 | 3 |
| SQ-CG | 90 | 7 | 2 | 3 |
| XC-SQ | 90 | 9 | 2 | 3 |
| ZZ-ZZN | 60 | 8 | 2 | 1 |
| ZZN-XLZ | 60 | 10 | 2 | 1 |
| XLZ-XZ | 60 | 10 | 2 | 1 |
| XZ-XD | 60 | 12 | 2 | 1 |
| XD-XZG | 60 | 14 | 2 | 1 |
| XZG-GT | 60 | 12 | 2 | 1 |
| GT-CG | 60 | 10 | 2 | 1 |
| CG-SQ | 60 | 10 | 2 | 1 |
| SQ-XC | 60 | 14 | 2 | 1 |
| ZZN-ZZ | 60 | 8 | 2 | 1 |
| XLZ-ZZN | 60 | 10 | 2 | 1 |
| XZ-XLZ | 60 | 10 | 2 | 1 |
| XD-XZ | 60 | 12 | 2 | 1 |
| XZG-XD | 60 | 14 | 2 | 1 |
| GT-XZG | 60 | 12 | 2 | 1 |
| CG-GT | 60 | 10 | 2 | 1 |
| SQ-CG | 60 | 10 | 2 | 1 |
| XC-SQ | 60 | 14 | 2 | 1 |

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
