# Peer review of "Rescheduling Out-of-Gauge Trains with Speed Restrictions and Temporal Blockades on the Opposite-Direction Track"

_mathematics, doi:10.3390/math11122659_

Round 1

Reviewer 1 Report

1. As the Author acknowledges, the weakest part of this research is "computational quality and efficiency". A suggestion in this regard is to verify if a relaxation of the model of page 3 by excluding equations (5) is a min-cost flow problem (and then it can be efficiently solved using, e.g., the network simplex). If this is true, perhaps combining the network simplex method with a  metaheuristic for (vertex) graph coloring (see, e.g., Wang and Xu, 2013) to incorporate equations (5) may be worth to try.

2. It is suggested to illustrate tables 3, 4 and 5 using appropriate graphs.

Bibliography

Wang, F., & Xu, Z. (2013). Metaheuristics for robust graph coloring. Journal of Heuristics19, 529-548.

A careful review is suggested. There are some typos that should be corrected, for example,

Line 182 on page 5, "discussed" may be more apprpriate than "discussing"

Line 200 on page 5, "which are" may be more apprpriate than "which is"

Line 220 on page 5, replace "fright" by "freight"

Line 388 on page 13, lines 391, 397, 403 on page 14. I believe "Constraints" or "Equations" my be more appropriate than "Formulation"

Line 443 on page 16, I believe "in C#" my be more appropriate than "by C#"

Reviewer 2 Report

The paper studies a train rescheduling problem when there are out-of-gauge trains (possibly adding additional speed restrictions or temporal blockades to itself and other trains). A combination of a time-space network with station tracks, a time-space resource table (for station track assignment), and a time-space speed allowance table (for available speed) is used to model the problem. Then an integer programming model of the problem is formed and used as a submodel in the solution method. The solution method uses a rolling time window horizon to create a subproblem which is solved as a typical integer programming problem. Then the condition on resource and speed allowance from the previously solved subproblems are used as a condition on the next time window subproblem.

The paper is very interesting. The overall presentation is very precise and easy to follow. I only have some questions/suggestions that may strengthen the paper. 

1. In Section 4, one series of datasets is generated for the experiment. However, the authors did not provide enough overall information about the dataset such as the number of trains of each type (passenger, freight, out-of-gauge), the number of stations, etc. The author can provide the full information or just a summary so that the dataset can be roughly reproduced. Also, the name of each instance should be briefly explained.

2. In Figure 12 and in the rest of Section 4, the terms Level 1 and Level 2 are used without proper definitions.

3. I understand that the paper has a main focus on proposing the problem formulation and the time horizon solution method, and has less focus on studying the effect of the parameters in the problem. However, I found a few conclusions that are now quite followed by the experiment result. (Line numbers are referred from the attached PDF file.)

3.1 The conclusion on Lines 521-522.

3.2 The conclusion on Lines 604-605.

4. In the paragraph beginning at Line 587, four strategies (IP, RS-P&F, RS-P&NOF&OF, RS-A) are considered. However, the result in Table 6 is not for these four strategies.

5. The lower time window gives a better result in Figure 12, while gives a worse result in Table 5. Some possible explanations would reduce the confusion.

6. Is the rescheduled timetable in Figure 14 the result of any solution method? Or is it available data prior to the study (such as a reschedule made by an expert)? In the latter case, we can compare the result from the solution methods to such data from the expert.

7. There are some minor errors/typos.

7.1 The reference (Liao et. al30) on Line 315.

7.2 The variable e(s1,s2) on Line 353 and the rest of the section has not been defined.

7.3 Section 0 on Line 380.

7.4 The multiplication symbol in Equation (1).

---

Reviewer 3 Report

1. Wikipedia is not accepted as reference.

2. Some references older than 2019 should be replaced with the newer one.

3. The most important quantitative result (total train delay reduction) should be presented in abstract.

4. There should be some citations in the first paragraph in introduction.

5. Background sub section lacks of citations.

6. Review regarding train scheduling/rescheduling problem should be summarized in a table to make it easier to read.

7. The references should be enriched with DOI wherever possible.

8. Literature review should also be enriched with studies regarding out-of-gauge train.

9. Conclusion lacks of quantitative result.

10. Conclusion should also be enriched with proposal for further studies.

11. There are some grammatical errors. For example: The computational efficiency of the problem "not" only depends on... (page 17).

12. Authors has presented the mathematical model through integer programming (objective and constraints). But, the algorithm used for the optimization has not been clearly presented.

13. The simulation scenario should be presented before the simulation result.

14. The competitor used for comparison in the simulation has not been presented clearly, including the reason of choosing the competitor.

14. The rationale of choosing the simulation screnario as in page 20 has not been presented clearly. 

Authors should fix the grammatical mistakes in this paper.

Reviewer 4 Report

1. Fig.3: please describe upper and  lower axis (the departure/arrival time or universal time axis ?); too long caption (phrase out-of-gauge repeats too many times); caption should start from the capital letter

2. too wide tables 3, 4, 6, 13, captions should start from the capital letter

3, Too wide figures 13, 14;  in 13 caption should start from the capital letter; fig 14 has too wide caption (please reduce)

4. Figs. 3, 5, 7, 9:, 11: caption should start from the capital letter;
